# Pasta Consumption and Connected Dietary Habits: Associations with Glucose Control, Adiposity Measures, and Cardiovascular Risk Factors in People with Type 2 Diabetes—TOSCA.IT Study

**DOI:** 10.3390/nu12010101

**Published:** 2019-12-30

**Authors:** Marilena Vitale, Maria Masulli, Angela Albarosa Rivellese, Enzo Bonora, Anna Carla Babini, Giovanni Sartore, Laura Corsi, Raffaella Buzzetti, Giuseppe Citro, Maria Pompea Antonia Baldassarre, Antonio Carlo Bossi, Carla Giordano, Stefania Auciello, Elisabetta Dall’Aglio, Rossella Iannarelli, Laura Tonutti, Michele Sacco, Graziano Di Cianni, Gennaro Clemente, Giovanna Gregori, Sara Grioni, Vittorio Krogh, Gabriele Riccardi, Olga Vaccaro

**Affiliations:** 1Department of Clinical Medicine and Surgery, University of Naples Federico II, 80131 Naples, Italy; marilena.vitale@unina.it (M.V.); maria.masulli@unina.it (M.M.); rivelles@unina.it (A.A.R.); astefaniamichela@gmail.com (S.A.); riccardi@unina.it (G.R.); 2Division of Endocrinology, Diabetes and Metabolism, University and Hospital Trust of Verona, 37138 Verona, Italy; enzo.bonora@univr.it; 3Medical Division, Rimini Hospital, 47900 Rimini, Italy; acbabini@auslrn.net; 4Dipartimento di Medicina, University of Padova, 35100 Padova, Italy; g.sartore@unipd.it; 5Dipartimento di Diabetologia e Malattie del Metabolismo, ASL 4 Chiavarese, 16043 Genova, Italy; lcorsi@asl4.liguria.it; 6Department of Experimental Medicine, Sapienza University, 04100 Rome, Italy; raffaella.buzzetti@uniroma1.it; 7UO Endocrinologia e Diabetologia, ASP, 85100 Potenza, Italy; giuseppe.citro@aspbasilicata.it; 8CeSI-MeT-Centro di Scienze dell’Invecchiamento e Medicina Traslazionale-University G. D’Annunzio of Chieti, 66100 Pescara, Italy; marbaldassarre@gmail.com; 9UOC Malattie Endocrine e Centro Regionale per il Diabete Mellito, ASST Bergamo Ovest di Treviglio, 24047 Treviglio, Italy; acbossi@gmail.com; 10Section of Endocrinology, Diabetology and Metabolic Diseases, University of Palermo, 90127 Palermo, Italy; carlagiordano53@gmail.com; 11Dipartimento di Medicina Clinica e Sperimentale, University of Parma, 43100 Parma, Italy; elisabetta.dallaglio@unipr.it; 12UOSD Diabetologia e Malattie del Metabolismo, Ospedale San Salvatore, 67100 L’Aquila, Italy; ross.iann29.03@gmail.com; 13SOC di Endocrinologia e Malattie del Metabolismo, AOU S. Maria della Misericordia, 33100 Udine, Italy; tonutti.laura@aoud.sanita.fvg.it; 14Center for Outcomes Research and Clinical Epidemiology (CORESEARCH), 65124 Pescara, Italy; sacco@coresearch.it; 15UOC Diabetologia, ASL 6, 57100 Livorno, Italy; graziano.dicianni@gmail.com; 16IRPPS Research National Council, Penta di Fisciano, 84025 Salerno, Italy; gennaro.clemente@cnr.it; 17SSD Diabetologia, ASL 1, 54100 Massa Carrara, Italy; g.gregori@usl1.toscana.it; 18Unità di Epidemiologia e Prevenzione, Fondazione IRCCS, Istituto Nazionale Tumori, 20133 Milano, Italy; Sara.Grioni@istitutotumori.mi.it (S.G.); Vittorio.Krogh@istitutotumori.mi.it (V.K.)

**Keywords:** type 2 diabetes, pasta consumption, dietary habits, glucose control, body mass index, cardiovascular risk factors

## Abstract

Background: Pasta is a refined carbohydrate with a low glycemic index. Whether pasta shares the metabolic advantages of other low glycemic index foods has not really been investigated. The aim of this study is to document, in people with type-2 diabetes, the consumption of pasta, the connected dietary habits, and the association with glucose control, measures of adiposity, and major cardiovascular risk factors. Methods: We studied 2562 participants. The dietary habits were assessed with the European Prospective Investigation into Cancer and Nutrition (EPIC) questionnaire. Sex-specific quartiles of pasta consumption were created in order to explore the study aims. Results: A higher pasta consumption was associated with a lower intake of proteins, total and saturated fat, cholesterol, added sugar, and fiber. Glucose control, body mass index, prevalence of obesity, and visceral obesity were not significantly different across the quartiles of pasta intake. No relation was found with LDL cholesterol and triglycerides, but there was an inverse relation with HDL-cholesterol. Systolic blood pressure increased with pasta consumption; but this relation was not confirmed after correction for confounders. Conclusions: In people with type-2 diabetes, the consumption of pasta, within the limits recommended for total carbohydrates intake, is not associated with worsening of glucose control, measures of adiposity, and major cardiovascular risk factors.

## 1. Introduction

Type-2 diabetes mellitus is a condition primarily defined by the level of hyperglycemia, and it is associated with a reduced life expectancy, significant morbidity due to specific diabetes related microvascular complications, increased risk of macrovascular complications (ischemic heart disease, stroke, and peripheral vascular disease), and diminished quality of life. It is a chronic condition requiring continuous medical care with multifactorial risk-reduction strategies beyond glycemic control. Lifestyle management is a fundamental aspect of diabetes care and includes diabetes self-management education, nutrition therapy, physical activity, smoking cessation counseling, and psychosocial care.

Regarding nutrition therapy, the main goal is to improve glucose control and the cardiovascular risk factors profile, and to reduce weight, when needed [1]. Nutritional recommendations have been issued by several scientific societies in order to support clinicians in the management of diabetes [2,3,4]. Although there is no full agreement about the optimal composition of the diet, cereal foods—particularly those with a high glycemic index—are generally restricted because of their adverse effects on post-prandial plasma glucose and lipids [4,5]. Cereal foods are a major source of energy and are largely consumed worldwide under different forms; but there is remarkably little information on the health impact of the specific food items consumed in different cultures. Pasta, a traditional Italian dish now largely available around the world, is an important example of a food that is considered to be a refined carbohydrate, but has a low–moderate glycemic index (GI) and a low caloric density, [6,7,8], and therefore may be a more suitable source of carbohydrates than other cereal foods for people with type-2 diabetes. Recently, the consumption of pasta has considerably declined, as the concept of a low carbohydrate and high protein diet against obesity emerged. The scientific evidence linking pasta consumption with obesity/overweight is however scarce. A recent metanalysis of randomized controlled trials has shown that in people with or without diabetes, pasta within the context of low-GI dietary patterns does not adversely affect adiposity; on the contrary, it slightly reduces body weight as compared with higher-GI dietary patterns [9]. These findings are in line with the results of observational studies showing benefits of pasta consumption on body weight and markers of adiposity [10,11]. In one of these studies, a greater pasta consumption was also associated with a better adherence to the Mediterranean diet [10]—a dietary pattern with proven beneficial effects on type-2 diabetes and cardiovascular risk [12].

As for the other relevant effects, a systematic review on the relation between pasta consumption and cardio-metabolic risk factors convincingly shows that pasta meals induce lower postprandial glucose responses compared with other starch-dense foods, like white bread and potatoes, commonly consumed in Western diets [13]. Most of the studies reviewed, however, are of a relatively short duration, and seldom include measurements of major cardiovascular risk factors, so that no conclusions on the effect of pasta consumption on major cardiovascular risk factors can be drawn. It remains therefore unclear to date whether, and to what extent, pasta shares the cardiometabolic advantages of other low GI foods in people with diabetes.

Furthermore, foods are consumed based on an exchange list principle (i.e., a larger consumption of some foods is often counterbalanced by a lesser consumption of other foods), and a greater consumption of pasta may be associated with a lower intake of energy-dense foods, rich in animal fat, which may adversely affect body weight, plasma lipids, and in turn cardiovascular health. Finally, as adherence to the nutritional recommendations in people with diabetes is generally low [14,15], the inclusion in the dietary plan of local, traditional dishes to be shared with other members of the community may considerably improve adherence.

The aim of this study is to document, in clinical practice, the consumption of pasta and the connected dietary habits in a cohort of people with type-2 diabetes, and to evaluate the association of pasta consumption with glucose control, measures of adiposity, and major cardiovascular risk factors.

## 2. Materials and Methods

We studied 2562 men and women with type-2 diabetes, aged 5–75 years, enrolled in the TOSCA.IT study, a randomized clinical trial (clinicaltrials.gov NCT00700856) designed to compare the impact of glucose-lowering drugs on cardiovascular events [16]. The study protocol was approved by the Ethics Review Committee of the coordinating center and of each participating center; written informed consent was obtained from all participants. People with cardiovascular events in the previous six months, cancer in the previous five years, or any disease, other than diabetes, requiring special dietary treatment(s), were excluded from the study. The details of the study protocol and the main results of the trial have been previously published [17]. For the purposes of the present analysis, we used data collected at study entry, prior to randomization to the study treatments. The data were analyzed according to a cross-sectional observational design.

The body weight and height, waist, and hip circumference were measured according to a standard protocol. The body mass index (BMI) was calculated as weight (kg)/height (m^2^). The percentage of people with abdominal obesity was also calculated according to the ATPIII criteria (waist circumference ≥102 cm for men and ≥88 cm for women). Biochemical analyses were performed in a central laboratory on fasting blood samples. HbA1c was measured with high-performance liquid chromatography (HPLC). The total cholesterol, HDL-cholesterol, and triglycerides were measured by standard methods. The LDL-cholesterol was calculated according to the Friedewald equation, only for people with triglycerides values <400 mg/dL. For people with triglyceride values >400 mg/dL (seven in total), the LDL cholesterol was considered as a missing value. By protocol, all patients were taking the same treatment for diabetes (i.e., metformin 2 g/day) [17]. Information on the concomitant treatments were collected by standard questionnaires.

### 2.1. Collection of Dietary Information

Dietary habits were investigated with the European Prospective Investigation into Cancer and Nutrition (EPIC) food frequency questionnaire, adapted for the Italian population [18,19]. The questionnaire contains 248 items, including the type of fat used as a condiment or added after cooking. The respondent indicates the absolute frequency of consumption of each item (per day, week, month, or year). The quantity of foods consumed was assessed based on the respondent’s selection of an image of a food portion showing a small, medium, or large portion size, with additional quantifiers (i.e., “smaller than the small portion” or “between the small and medium portion”). Software developed ad hoc by the Epidemiology and Prevention Unit, Fondazione IRCCS, Istituto Nazionale dei Tumori, Milan, was used to convert the questionnaire dietary data into frequencies of consumption (times per day) and average daily amounts of foods (grams per day). The energy intake (kcal per day) and nutrient composition of the habitual diet were estimated by the software, through the link of the consumed foods to the Italian Food Composition Tables [20,21] for energy and nutrients assessment. The glycemic index was calculated according to the tables by Brighenti [7], and the International Table by Foster and Powell [8]. Incomplete questionnaires and questionnaires with implausible data were excluded from the analyses. The intake of macronutrients is given as a percentage of the total energy or g/1000 Kcal/day, or an absolute value, as appropriate.

### 2.2. Statistical Analysis

The data are expressed as means and standard deviations, or frequencies and percentages, as appropriate. To correct for the different energy intake between subjects, the intake of pasta was expressed as g/1000 Kcal/day. The relation of pasta consumption with anthropometric variables, blood pressure, glucose control, and plasma lipids, was explored by stratifying the study population according to quartiles of pasta consumption; in order to account for gender differences in pasta consumption, allocation into quartiles was based on the sex-specific distribution of the amount consumed. To compare variables across the quartiles, an analysis of variance with a test for a linear trend was used for the continuous variables, and the χ^2^ test was used to compare proportions. The multivariate linear regression model was used to evaluate the separate and combined association of pasta and other covariates with the variables of interest. Covariates were age, gender, BMI, smoking, fiber, and alcohol intake. The statistical analyses were performed with SPSS version 19.0. Statistical significance was assessed as *p* < 0.05 (two-sided).

## 3. Results

The study population consisted of 1531 men and 1031 women with type-2 diabetes. The average age was 62.1 ± 6.5 years and the average BMI was 30.3 ± 4.4 Kg/m2. The average pasta consumption was 53.0 ± 40.4 g/day, and was higher in men than in women (*p* < 0.05). The clinical and socio-demographic characteristics of the study population are given in Table 1, according to quartiles of pasta consumption. Age, diabetes duration, smoking habits, educational status, and geographical area of residence were not significantly different across the categories of pasta consumption.

Table 2 describes the composition of the diet. The average daily energy intake was similar across the quartiles of pasta consumption, but the nutrient composition of the diet was significantly different. A higher pasta consumption was associated with a significantly higher intake of total carbohydrates, significantly lower intake of proteins, total fat, saturated fat, unsaturated fat, cholesterol, added sugar, alcohol, and fiber. The glycemic load of the diet was not significantly different across quartiles of pasta consumption, whereas the glycemic index was inversely associated with pasta consumption.

A higher pasta consumption was also associated with a better adherence to the nutritional recommendations for people with diabetes for all items except for fiber, which was inversely associated with pasta consumption (Figure 1).

Glucose control, evaluated by glycated hemoglobin, did not change significantly across the quartiles of pasta consumption (Table 3). In this regard, it is relevant to say that because of the eligibility criteria of the trial, the study participants were all taking the same hypoglycemic treatment (i.e., metformin 2 g/day). No relation was observed between pasta consumption and measures of adiposity. BMI, prevalence of obesity (BMI ≥30 Kg/m2), and surrogate measures of visceral adiposity—such as waist circumference, waist/hip ratio, and prevalence of abdominal obesity—were not significantly different across the categories of pasta consumption (Table 3). As for major cardiovascular risk factors, non-significant differences were observed for LDL-cholesterol and triglycerides, whereas HDL-cholesterol was significantly and inversely associated with pasta consumption. The LDL to HDL ratio and the non-HDL-cholesterol, two markers of the atherogenicity of the plasma lipids profile, remained stable across the quartile of pasta consumption. Systolic blood pressure was significantly and directly associated with pasta consumption, whereas no difference was detected for diastolic blood pressure. A high proportion of the cohort was taking lipid lowering medications (61.2%) and antihypertensive medications (92.3%), with no significant differences across categories of pasta consumption (Table 3).

To disentangle the association of pasta per se with HDL-cholesterol and systolic blood pressure, we performed a multivariate regression analysis with age, gender, BMI, smoking, fiber, and alcohol intake as covariates. Pasta consumption remained significantly and inversely associated with HDL-cholesterol, but not with systolic blood pressure (Table 4).

## 4. Discussion

The study provides evidence that in people with type-2 diabetes, the consumption of cereals in the form of pasta, not exceeding the limits recommended for total carbohydrate intake, is associated with a lower intake of proteins, total and saturated fat, cholesterol, and added sugar; this increases the overall adherence to the nutritional recommendation for the management of diabetes, which emphasizes the replacement of foods high in fat and sugar, with carbohydrates low in energy density and with a low glycemic index [1,2,3,22]. Furthermore, pasta consumption is not associated with a worsening of glucose control, BMI, obesity, abdominal adiposity, and the overall atherogenicity of the plasma lipids profile. However, a higher pasta consumption is associated with a lower intake of fiber, which makes the public health message regarding the consumption of pasta more complex than previously considered. Pasta is, in fact, a rare example of a refined carbohydrate food with a low GI.

The beneficial effects of low GI foods in people with and without diabetes have been established [23], but low GI diets can be achieved through different ways. The low GI of pasta is mainly due to its dense structure, resulting in slower digestion and delayed gastric emptying, which makes pasta a unique example of a refined cereal food with a low GI. To what extent pasta shares the positive metabolic effects of other low GI foods with a high fiber content remains unclear [24,25]. A recent meta-analysis convincingly shows that, in relatively short-term controlled studies, pasta meals induce lower postprandial glucose and possibly lower insulin response, compared with other carbohydrate-dense foods such as white bread or potatoes—widely consumed in Western diets [13]. The results of our study expand the current knowledge by showing that in people with type 2 diabetes, habitual pasta consumption, not exceeding the limits recommended for total carbohydrate intake is not associated with a worsening of glycated hemoglobin, a marker of long-term glucose control and a significant predictor of the development of diabetes complications [26]. Furthermore, we show that pasta consumption is not associated with a higher BMI, prevalence of obesity, or abdominal adiposity. These findings do not lend support to the commonly held paradigm that pasta is “fattening”. In contrast with this misconception and in line with our results, a recent meta-analysis of randomized controlled studies has shown that in people with or without diabetes, pasta consumption, within the context of a low GI eating pattern, is associated with a lower BMI compared with high GI eating patterns [9]. Along the same line are the results of two observational studies, which demonstrate the benefits of pasta consumption on body weight and adiposity in large, free-living populations [10,11].

Unlike prior studies, we have explored the associations between pasta consumption and major cardiovascular risk factors. An unwanted effect of carbohydrate intake on HDL-cholesterol has been described previously [27]. In line with these findings, we observe a significant inverse association of pasta consumption with HDL-cholesterol; although statistically significant, the magnitude of the change was of moderate clinical relevance. Furthermore, because of a concomitant, non-significant reduction in LDL-cholesterol, in this study, the ratio LDL to HDL cholesterol, and the values of non-HDL cholesterol did not change across the pasta quartiles. This is relevant in that non-HDL cholesterol represents a good estimation of the total amount of potentially atherogenic apolipoprotein B-containing lipid fractions (i.e., LDL, lipoprotein-a, intermediate-density lipoprotein, very low-density lipoprotein, and VLDL remnants), and has a well-established role as an independent predictor of cardiovascular disease, even in patients with normal LDL and triglyceride levels [28]. Non-HDL-cholesterol is, in fact, recommended as a secondary lipid target when the LDL-cholesterol goal is reached, particularly in people with insulin resistance—a common feature of type-2 diabetes [29].

We found a direct association of pasta consumption with systolic blood pressure, which was, however, largely mediated by major confounders such as age and BMI. The influence of dietary carbohydrates on blood pressure is controversial—both the quality and quantity of carbohydrates may play a role [30]. To our knowledge, the relationship of pasta consumption with blood pressure has not been explored before. The present study is the first exploring, in real life clinical practice, the eating habits and quality of the diet associated with pasta consumption in people with type-2 diabetes, with a focus on the cardio-metabolic profile.

A major study strength is that the intake of food and nutrients reflects real life conditions. Furthermore, the use of a central laboratory for biochemical analyses ensures good quality standards for the measurements.

Among the limitations, we acknowledge the cross-sectional design, which may limit the assessment of a cause–effect relationship. Reverse causation is a concern when the relation of dietary habits with cardio-metabolic markers and adiposity measures is assessed cross-sectionally. The individual’s perception of cardio-metabolic and weight status may influence both the dietary habits and their reporting, and may produce spurious associations. In order to limit the possibility of spurious associations, we have excluded from the analysis implausible dietary questionnaires and the people with acute cardiovascular events in the prior six months. Furthermore, we observed a significant and positive association between BMI and energy intake in the whole population, which supports the internal validity of the data. Additional problems are the lack of information on wholegrain cereal intake, their consumption, however, is rather limited; a previous study conducted in an Italian cohort has shown that the average consumption of wholegrain cereals is about 5 g per day [31]. In addition, the dietary data were collected only once, and could be prone to recall bias and seasonal variation. Finally, the data collected do not allow us to analyze the different types of pasta dishes. This may be relevant, as both healthy and unhealthy eating patterns have been described in pasta eaters [32], and this may depend on the way pasta dishes are prepared.

Regarding the study population, we cannot exclude that the use of hypoglycemic medications, as well as the extensive use of lipid lowering and anti-hypertensive drugs, may have partly masked a possible association of pasta with worsening of glucose control and cardiovascular risk factors. It is however relevant in this regard to underline that the use of medications was evenly distributed across the categories of pasta consumption.

Despite these, the study findings are relevant, in as much as the study population represents most of the patients with type-2 diabetes encountered in clinical practice.

## 5. Conclusions

In conclusion, in people with type-2 diabetes, the consumption of cereal foods in the form of pasta, not exceeding the limits recommended for total carbohydrate intake is not associated with the worsening of glucose control, BMI, adiposity measures, obesity, and major cardiovascular risk factors. If these results are confirmed, pasta may represent a suitable food choice to widen the dietary option in people with diabetes, particularly when consumed in dishes that combine pasta with vegetables and legumes, according to the Mediterranean traditional diet.

## Figures and Tables

**Figure 1 nutrients-12-00101-f001:**
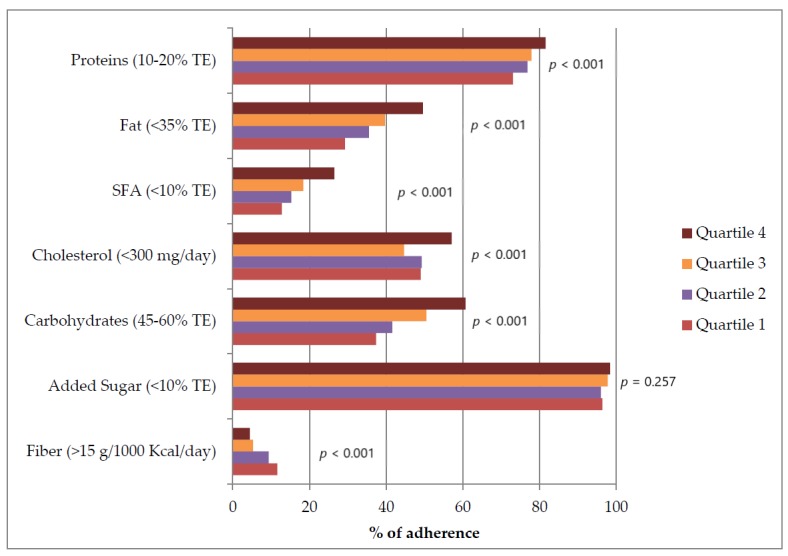
Adherence to the nutritional recommendations [2,3] by sex-specific quartiles of pasta consumption, expressed as g/1000 Kcal. TE—total energy; SFA—saturated fatty acids.

**Table 1 nutrients-12-00101-t001:** Characteristics of the study population by quartiles of pasta consumption expressed as (g/1000 Kcal).

	Quartile 1 (*n* = 640)	Quartile 2 (*n* = 641)	Quartile 3 (*n* = 641)	Quartile 4 (*n* = 640)	*p*-Value
Pasta consumption (g/1000 Kcal)	7.5 ± 4.4 (0.0–15.8)	19.4 ± 4.2 (15.9–27.4)	30.8 ± 5.0 (27.5–41.1)	52.4 ± 14.0 (41.2–120.0)	
Age (years)	62.0 ± 6.5	62.3 ± 6.5	62.0 ± 6.5	62.1 ± 6.4	0.842
Diabetes Duration (years)	8.5 ± 5.8	8.3 ± 5.4	8.7 ± 6.0	8.5 ± 5.6	0.658
Smoking status (%) ^a^					
Current Smokers	125 (19.5)	114 (17.8)	97 (15.1)	106 (16.6)	0.339
Educational status (%)					
None	194 (30.4)	197 (30.7)	180 (28.1)	218 (34.1)	0.204
Primary	226 (35.4)	248 (38.6)	250 (39.0)	231 (36.1)	
Secondary	185 (29.0)	160 (24.9)	174 (27.1)	169 (26.4)	
University	34 (5.3)	36 (5.8)	37 (5.8)	22 (3.4)	
Geographical area (%)					
North	224 (34.9)	216 (33.8)	218 (34.0)	233 (36.4)	0.681
Center	156 (24.6)	182 (28.3)	176 (27.5)	159 (24.8)	
South	260 (40.5)	243 (37.9)	247 (38.5)	248 (38.8)	

^a^ Subjects were classified as “current smokers” if they smoked five or more cigarettes/day.

**Table 2 nutrients-12-00101-t002:** Total energy and macronutrient intake in the study population by quartiles of pasta consumption (g/1000 Kcal).

	Quartile 1 (*n* = 640)	Quartile 2 (*n* = 641)	Quartile 3 (*n* = 641)	Quartile 4 (*n* = 640)	*p* for Anova	*p* for Trend
Pasta (g/1000 Kcal)	7.5 ± 4.4 (0.0–15.8)	19.4 ± 4.2 (15.9–27.4)	30.8 ± 5.0 (27.5–41.1)	52.4 ± 14.0 (41.2–120.0)		
Energy (Kcal)	1841 ± 820	1962 ± 881	1941 ± 683	1852 ± 673	0.067	0.519
Protein (% of TE)	18.6 ± 2.8	18.3 ± 2.5	18.3 ± 2.4	17.9 ± 2.5	<0.0001	<0.0001
Total Fat (% of TE)	38.3 ± 6.4	37.1 ± 6.1	36.3 ± 5.8	35.1 ± 5.7	<0.0001	<0.0001
SFA (% of TE)	12.8 ± 2.7	12.4 ± 2.5	12.1 ± 2.4	11.4 ± 2.3	<0.0001	<0.0001
MUFA (% of TE)	18.4 ± 4.1	18.0 ± 3.9	17.8 ± 3.6	17.5 ± 3.7	0.002	<0.0001
PUFA (% of TE)	4.7 ± 1.3	4.5 ± 1.0	4.3 ± 1.0	4.2 ± 1.0	<0.0001	<0.0001
Cholesterol (mg/1000 Kcal)	191 ± 54	183 ± 54	180 ± 50	172 ± 49	<0.0001	<0.0001
Carbohydrates (% of TE)	43.1 ± 7.7	44.6 ± 7.5	45.4 ± 7.0	46.9 ± 6.8	<0.0001	<0.0001
Added Sugar (% of TE)	2.7 ± 3.7	2.4 ± 3.3	2.3 ± 3.2	1.9 ± 2.5	<0.0001	<0.0001
Fiber (g/1000 Kcal)	11.5 ± 3.0	11.0 ± 2.9	10.5 ± 2.6	10.1 ± 2.4	<0.0001	<0.0001
Glycemic index (%)	53.0 ± 3.6	52.4 ± 3.2	51.5 ± 2.9	49.7 ± 2.9	<0.0001	<0.0001
Glycemic load	111 ± 64	118 ± 65	119 ± 50	111 ± 46	0.078	0.341
Alcohol (g/day)	10.7 ± 15.9	12.2 ± 18.4	11.2 ± 14.6	9.1 ± 13.8	0.046	0.023

TE—total energy; SFA—saturated fatty acids; MUFA—monounsaturated fatty acids; PUFA—polyunsaturated fatty acids.

**Table 3 nutrients-12-00101-t003:** Clinical characteristics of the study population by quartiles of pasta consumption expressed as g/1000 Kcal.

	Quartile 1 (*n* = 640)	Quartile 2 (*n* = 641)	Quartile 3 (*n* = 641)	Quartile 4 (*n* = 640)	*p* for Anova or *χ*^2^	*p* for Trend
Pasta (g/1000 Kcal)	7.5 ± 4.4 (0.0–15.8)	19.4 ± 4.2 (15.9–27.4)	30.8 ± 5.0 (27.5–41.1)	52.4 ± 14.0 (41.2–120.0)		
BMI (Kg/m^2^)	30.1 ± 4.4	30.5 ± 4.5	30.2 ± 4.5	30.4 ± 4.4	0.362	0.569
BMI ≥30 Kg/m^2^ (%)	48.6	49.5	48.0	49.7	0.926	
Waist Circumference (cm)	103.9 ± 11.2	104.8 ± 11.1	104.7 ± 11.2	104.4 ± 11.3	0.497	0.524
Waist-to-hip ratio	0.98 ± 0.07	0.98 ± 0.06	0.98 ± 0.07	0.98 ± 0.07	0.545	0.154
With Abdominal obesity * (%)	71.9	74.6	75.0	74.1	0.589	
HbA1c (%)	7.67 ± 0.51	7.70 ± 0.50	7.67 ± 0.51	7.68 ± 0.51	0.707	0.915
Proportion with HbA1c <7.5%	26.0	23.7	25.8	24.4	0.445	
LDL-cholesterol (mg/dl)	104.7 ± 32.3	102.3 ± 31.5	102.2 ± 31.6	102.0 ± 30.3	0.364	0.142
HDL-cholesterol (mg/dl)	47.8 ± 12.5	45.5 ± 11.8	46.1 ± 11.9	44.9 ± 11.3	<0.0001	<0.0001
LDL/HDL Ratio	2.32 ± 0.90	2.36 ± 0.90	2.33 ± 0.90	2.39 ± 0.92	0.571	0.271
Triglycerides (mg/dl)	145.7 ± 71.5	153.4 ± 80.2	148.7 ± 72.6	155.2 ± 75.4	0.095	0.072
Non-HDL Cholesterol (mg/dl)	133.8 ± 36.0	133.3 ± 36.3	132.9 ± 37.9	133.2 ± 35.0	0.980	0.963
Systolic blood pressure (mm/Hg)	132.5 ± 14.3	133.7 ± 13.9	135.2 ± 15.2	133.8 ± 15.0	0.034	0.012
Diastolic blood pressure (mm/Hg)	78.9 ± 8.8	79.8 ± 8.6	80.1 ± 8.3	79.4 ± 8.4	0.103	0.300
On lipid lowering medications (%)	59.7	66.1	62.8	59.9	0.080	
On antihypertensive medications (%)	93.6	90.3	94.1	92.0	0.104	

* Waist circumference: >102 cm for men and >88 cm for women. BMI—body mass index.

**Table 4 nutrients-12-00101-t004:** Multivariate regression analyses of the relation between pasta consumption and HDL-cholesterol and systolic blood pressure.

	Standardized ß-Coefficient	Standard Error	*p*-Value
Dependent variable HDL-cholesterol			
Intake of pasta (g/1000 Kcal)	−0.057	0.007	0.003
Gender (female/male)	0.330	0.535	0.000
BMI (Kg/m^2^)	−0.170	0.050	0.000
Age (years)	0.093	0.034	0.000
Fiber intake (g/1000 Kcal)	0.004	0.032	0.836
Alcohol (g/day)	0.171	0.015	0.000
Smoking (yes/no)	−0.018	0.266	0.370
Dependent variable Systolic Blood Pressure			
Intake of pasta (g/1000 Kcal)	0.031	0.010	0.133
Gender (female/ male)	−0.026	0.833	0.261
BMI (Kg/m^2^)	0.108	0.078	0.000
Age (years)	0.126	0.054	0.000
Fiber intake (g/1000 Kcal)	−0.045	0.049	0.024
Alcohol (g/day)	0.001	0.023	0.970
Smoking (yes/no)	0.044	0.415	0.040

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
