# Peer review of "Pasta Consumption and Connected Dietary Habits: Associations with Glucose Control, Adiposity Measures, and Cardiovascular Risk Factors in People with Type 2 Diabetes—TOSCA.IT Study"

_nutrients, 2019, doi:10.3390/nu12010101_

Round 1
Reviewer 1 Report
Overall, the manuscript is fine. I only have some suggestions as below.
It will be good to add a paragraph in the introduction section to briefly explain what is diabetes and why dietary management of diabetes is important in today's society. According to my website checks, raw data for the study has not been deposited in clinicaltrials.gov or any data repository yet. I will recommend the authors to deposit the raw data online before official publication of the analyzed results. The manuscript requires some proofreading and editing for its English and formating. e.g. line 90 "foods" should be "food", line 112 "m2" should be "m2", etc. I will not list all the minor mistakes here.
Author Response
Overall, the manuscript is fine. I only have some suggestions as below. It will be good to add a paragraph in the introduction section to briefly explain what diabetes and why dietary management of diabetes is important in today's society.
This paragraph has been added as suggested (page 2, lines 61-68).
According to my website checks, raw data for the study has not been deposited in clinicaltrials.gov or any data repository yet. I will recommend the authors to deposit the raw data online before official publication of the analyzed results.
We thank the reviewer for this suggestion. We agree that it is very important to deposit the raw data online, and we’ll provide as soon as possible. Anyway, the protocol of the trial is registered in clinicaltrials.gov, so row data to be deposited will refer to the trial results. The results reported in the present manuscript are based on data collected prior to the randomization to study treatments and are not part of the trial end points. This has been made clear (page 3, lines 117-119)
The manuscript requires some proofreading and editing for its English and formating. e.g. line 90 "foods" should be "food", line 112 "m2" should be "m2", etc. I will not list all the minor mistakes here.
We have done our best to correct all the typos and a linguistic revision was performed.
Reviewer 2 Report
Vitale et al. examined the impact of increased pasta consumption on glucose homeostasis, adiposity, and on common cardiovascular risk factors in people with type 2 diabetes. They find that increased pasta consumption lowers protein, total fat, saturated fat, cholesterol (including HDL cholesterol), and fiber intake. The study is well powered. However, the statistical approach should be described in detail. Some aspects require improvements.
In the Methods section (line 116) The LDL-cholesterol was calculated using the Friedewald equation for people with triglyceride levels of <400mg/dl. Please clarify what was used to calculate LDL levels for people that had higher TG levels? What were the steps taken to reconcile the two methods?
For all the regression analyses, was the log transformation of the Lipid variables performed?
Table 1 is sex-specific quartiles: Which sex is presented? Where is the data for the other sex?
Table 2 P for trend: Please explain how is that P-value is calculated.
Figure 1 and Table 1 don’t match. Decreased protein consumption with increased pasta intake but better adherence to nutritional recommendations? How is the adherence to nutritional recommendations assessed? What is the criteria for 100% adherence? How is the adherence percentage calculated ?
While the effect on HDL cholesterol is significant it is very modest. The clinical relevance of such effect should be discussed.
The authors should discuss the shortcomings of food frequency questionnaires in nutrition studies.
Author Response
Vitale et al. examined the impact of increased pasta consumption on glucose homeostasis, adiposity, and on common cardiovascular risk factors in people with type 2 diabetes. They find that increased pasta consumption lowers protein, total fat, saturated fat, cholesterol (including HDL cholesterol), and fiber intake. The study is well powered. However, the statistical approach should be described in detail. Some aspects require improvements.
The statistical methods have been described more clearly, please see text page 4.
In the Methods section (line 116) The LDL-cholesterol was calculated using the Friedewald equation for people with triglyceride levels of <400mg/dl. Please clarify what was used to calculate LDL levels for people that had higher TG levels? What were the steps taken to reconcile the two methods?
For people with triglyceride values >400 mg/dl (seven in total), LDL-cholesterol could not be calculated, and these people have been excluded from the analyses. This has been reported in the methods section (page 3, lines 127-128).
For all the regression analyses, was the log transformation of the Lipid variables performed?
The distribution of the lipid variables was tested for deviation from normality by calculating the Skewness of the distribution. The values ranged between -0.5 and 0.5, thus suggesting that the log transformation is not necessary.
Table 1 is sex-specific quartiles: Which sex is presented? Where is the data for the other sex?
Table 1 shows data for the whole population. Allocation into quartiles was based on sex-specific distribution of pasta consumption to account for the gender differences in pasta consumption. This has been clarified in the methods section (page 4, lines 153-154). To avoid confusion, we have deleted the term sex-specific quartiles from the titles of tables 1, 2 and 3.
Table 2 P for trend: Please explain how is that P-value is calculated.
For continuous variables, the analysis of variance (ANOVA) for weighted linear trend test was performed. This has been clarified in the methods section (statistical analysis) (page 4, lines 155-156).
Figure 1 and Table 1 don’t match. Decreased protein consumption with increased pasta intake but better adherence to nutritional recommendations?
Figure 1 and Table 1 report different analyses, they should not necessarily match. In particular, for proteins, the consumption generally exceeds the recommended intake, therefore, a lower protein intake translates into a greater adherence to the recommendations.
How is the adherence to nutritional recommendations assessed? What is the criteria for 100% adherence? How is the adherence percentage calculated?
Adherence to the nutritional recommendations has been assessed as follows: for each item (i.e. protein, fat, saturated fat, cholesterol, carbohydrate, added sugar, and fiber intake) a value of “1” was assigned to people whose consumption falls within the recommended range; a value of “0”was assigned to people who consumed less or more than the recommended intake. The percentage of adherence was calculated for each item as the number of people with value “1” / the total population * 100. For each item a 100% adherence means that all participants match the recommended intake.
While the effect on HDL cholesterol is significant it is very modest. The clinical relevance of such effect should be discussed.
This has been discussed as suggested (page 8, lines 246-250).
The authors should discuss the shortcomings of food frequency questionnaires in nutrition studies.
This has been discussed as suggested (page 9, lines 278-282).
Reviewer 3 Report
In this manuscript by Vitale el al, the authors aimed to establish the correlations of pasta consumption with its effects on glucose control, in type 2 diabetes patients. This study is well designed and provides inportant information regarding disease control using dietary recommendations. However, some critical concerns need to be carefully addressed before this manuscript is in a publishable fashion. Specific comments are listed below:
1. As the authors described in the limitation section, use of metformin may mask the effects of pasta consumption. As the data shown in Figure 3, about half of the study subjects are obese (BMI>30), however, their average cholesterol and blood pressure are within a normal range (maybe slightly higher). Is it an effect from the medication? If so, the authors should specifically discuss.
2. When calculating the protein and fiber content of the diet, does it include the content in the pasta? This may change the idea of how much non-pasta is consumed.
3. How is the glycemic load defined here? The numbers seem a bit high for my knowledge.
4. Why do saturated fat and fiber recommendations have the lowest adhesion rates among all other dietary recommendations? Can the authors discuss?
5. The manuscript has a few typos that need to be fixed. For example, in line 145, "or the 2 test"; line 218, "withe bread'.
Author Response
In this manuscript by Vitale el al, the authors aimed to establish the correlations of pasta consumption with its effects on glucose control, in type 2 diabetes patients. This study is well designed and provides inportant information regarding disease control using dietary recommendations. However, some critical concerns need to be carefully addressed before this manuscript is in a publishable fashion.
We thank the reviewer for the comment.
Specific comments are listed below:
As the authors described in the limitation section, use of metformin may mask the effects of pasta consumption. As the data shown in Figure 3, about half of the study subjects are obese (BMI>30), however, their average cholesterol and blood pressure are within a normal range (maybe slightly higher). Is it an effect from the medication? If so, the authors should specifically discuss.
We agree with the reviewer that this may be partly an effect of the medications, however the use of medications was evenly distributed across categories of pasta consumption, suggesting that this may not be a major confounder. This has been discussed in the text (page 9, line 283-287).
When calculating the protein and fiber content of the diet, does it include the content in the pasta? This may change the idea of how much non-pasta is consumed.
Yes, the protein and fiber content of the diet includes the content in the pasta.
How is the glycemic load defined here? The numbers seem a bit high for my knowledge.
The glycemic load has been calculated as: (GI * carbohydrate intake) / 100. We double checked the calculation and did not find any mistake.
Why do saturated fat and fiber recommendations have the lowest adhesion rates among all other dietary recommendations? Can the authors discuss?
Among others, this can be explained by the fact that generally people with diabetes tend to limit carbohydrate consumption and, therefore, increase the intake of proteins and lipids. In fact, foods and beverages are generally consumed based on exchange list principles (i.e. a lower consumption of some foods is often counterbalanced by a greater consumption of other foods). Furthermore, as foods rich in proteins and fats are generally low in fiber; consequently, the intake of fiber decreases. In any case, the findings we report are coherent with data collected in another cohort with diabetes (references 15) and this provides external consistency to the data.
The manuscript has a few typos that need to be fixed. For example, in line 145, "or the 2 test"; line 218, "withe bread'.
We have done our best to correct all the typos and a linguistic revision was performed.
Reviewer 4 Report
This was an interesting report on pasta consumption and the impact on obesity related risk factors. I’ve included some suggestions.
The mean intake was ~1/3 cup of pasta, which isn't a large amount considering the serving size of most pasta is 1/2 cup cooked. The range in Quartile 4 was also much larger than the other quartiles. Have you considered removing a possible outlier that may be skewing the range and rerunning analyses?
2. Protein, fat, fiber, and GI were lower in pasta consumers. What else were pasta consumers eating? Lower fiber but higher carb suggests refined CHO sources. Would expand upon this in discussion.
3. Pasta is considered a low to moderate GI food according to 2008 international GI tables. Would include a reference other than the international pasta organization for reference to low GI and low caloric density.
Author Response
This was an interesting report on pasta consumption and the impact on obesity related risk factors. I’ve included some suggestions.
We thank the reviewer for the comment.
The mean intake was ~1/3 cup of pasta, which isn't a large amount considering the serving size of most pasta is 1/2 cup cooked. The range in Quartile 4 was also much larger than the other quartiles. Have you considered removing a possible outlier that may be skewing the range and rerunning analyses?
As reported in the methods section, incomplete questionnaires and questionnaires with implausible data were excluded from the analyses. Having done that, we are fairly confident that the wide range in quartile 4 is not due to the inclusion of the outliers. On the other hand, the range of pasta consumption in quartile 4 is not unusual for Italian people.
Protein, fat, fiber, and GI were lower in pasta consumers. What else were pasta consumers eating?
Foods and beverages are generally consumed based on exchange list principles (i.e. a lower consumption of some foods is often counterbalanced by a greater consumption of other foods). It is, therefore, plausible that a higher consumption of pasta is associated with lower consumption of other food sources of proteins and fats.
Lower fiber but higher carb suggests refined CHO sources. Would expand upon this in discussion.
This has been discussed in the text (page 8, lines 223-224).
Pasta is considered a low to moderate GI food according to 2008 international GI tables. Would include a reference other than the international pasta organization for reference to low GI and low caloric density.
Two more references have been added.
Round 2
Reviewer 3 Report
The reviewer does not have further questions.
Author Response
A linguistic revision was performed.